# Neutralization and Improvement of Bauxite Residue by Saline-Alkali Tolerant Bacteria

**DOI:** 10.3390/ijerph191811590

**Published:** 2022-09-14

**Authors:** Lv Lv, Kunyan Qiu, Shiji Ge, Zhiqiang Jiao, Chenyang Gao, Haiguang Fu, Rongkui Su, Zhongkai Liu, Yulong Wang, Yangyang Wang

**Affiliations:** 1National Demonstration Center for Environmental and Planning, College of Geography and Environmental Science, Henan University, Kaifeng 475004, China; 2Key Laboratory of Geospatial Technology for the Middle and Lower Yellow River Regions, Henan University, Ministry of Education, Kaifeng 475004, China; 3Henan Engineering Research Center for Control & Remediation of Soil Heavy Metal Pollution, Henan University, Kaifeng 475004, China; 4Henan Key Laboratory for Monitoring and Remediation in Heavy Metal Polluted Soil, Jiyuan 459000, China; 5School of Environmental Science and Engineering, Central South University of Forestry and Technology, Changsha 410004, China; 6Zhengzhou Non-Ferrous Metals Research Institute Co., Ltd. of CHALCO, Zhengzhou 450041, China

**Keywords:** bauxite residue, neutralization, improvement, bacteria, *Bacillus* sp.

## Abstract

The high salt-alkalinity of bauxite residue (BR) hinders plant growth and revegetation of bauxite residue disposal areas (BRDA), which cause serious potential environmental and ecological risks. Bioneutralization is a promising method for improving the properties of BR and plant colonization. In the present study, a strong saline-alkali tolerant bacteria (ZH-1) was isolated from aged BR and identified as *Bacillus* sp. The medium of ZH-1 was optimized by orthogonal tests, and ZH-1 could decrease the medium pH from 11.8 to 6.01 (agitated culture) and 6.48 (static culture) by secretion of citric acid, oxalic acid and tartaric acid. With the inoculation of ZH-1, the pH of BR decreased from 11.6 to 8.76, and the water-soluble salt in BR increased by 68.11%. ZH-1 also changed the aggregate size distribution of BR, the mechanical-stable aggregates and water-stable aggregates increased by 18.76% and 10.83%, respectively. At the same time, the stability of the aggregates obviously increased and the destruction rate decreased from 94.37% to 73.46%. In addition, the microbial biomass carbon increased from 425 to 2794 mg/kg with the inoculation of ZH-1. Bacterial community analysis revealed that Clostridia, Bacilli, Gammaproteobacteria, Betaproteobacteria and Alphaproteobacteria were the main classes in the naturalized BR, and the inoculation of ZH-1 increased the diversity of bacteria in the BR. Overall, ZH-1 has great potential for neutralization and improvement the properties of BR and may be greatly beneficial for the revegetation of BRDA.

## 1. Introduction

Bauxite residue (BR) is an alkaline and saline bauxite processing waste, which is generated in the process of alumina production [1,2,3]. About 2 tons of BR is generated when 1 ton of alumina is produced, and then the BR are pumped to bauxite residue disposal areas (BRDA) [4,5]. At present, the global storage of BR has reached 3.5 billion tons and continues to increase by approximately 120 million tons per year [6]. The deposition of BR not only increases the salinity and alkalinity of surround environment [1], but it also has the risk to spill, especially in the rainy season, which poses a great threat to surrounding farmland and residents [7]. Therefore, great attention should be paid to the safety of BRDA.

Revegetation is considered to be the most promising method for the effective management of closed-down BRDA [8,9,10]. However, the physical and chemical deficiencies of BR properties seriously limit the colonization and growth of plants [8,11]. Thus, many researchers have focused on decreasing the pH and salinity and increasing the aggregation stability of BR, which can transfer BR to a soil-like material to promote the colonization and growth of plants [10]. Gypsum was the most commonly used amendment to remediate BR, which can reduce the pH and increase the leaching of Na^+^ from BR [12,13,14]. In addition, acid [15], biosolids, spent mushroom compost, green waste compost, biochar [8], sea water [16], carbon dioxide [17] and acidic fly ash [18] have also been used to reduce the pH and alkalinity of BR. However, for these methods the cost, need for complex equipment and supply of raw materials limit their large-scale application.

The development of biotechnology provides a new possibility for the remediation of BR. Several bacterial strains have been isolated from BR and used to remediate BR [19,20]. However, Santini et al. [2] reported that microbial-based strategies have not been trialed for BR remediation. After that, they tried to remediate BR with diverse microbial strains, and their result indicated that the bioremediation efficacy was mainly affected by the initial pH of BR and organic carbon dose [10]. In addition, an acid producing bacteria has been isolated and used to remediate BR [20], but they mainly emphasized their effect on the pH of BR. Previous reports have indicated that inoculation of functional bacteria can significantly improve the properties of soil [21,22,23]. Therefore, we hypothesize that inoculation of saline-alkali tolerant bacteria and organic matter (culture medium) can not only reduce the pH of BR but significantly improve its other properties as well.

In the present study, a saline-alkali tolerant bacterium was isolated and used to remediate and improve BR. Its influence on pH, total soluble salt, aggregation distribution and stability, and the morphology of BR were examined in detail. The survival of the bacteria and their diversity in BR were also examined by Illumina sequencing technology. The results of the present study can provide alternative technology for the remediation of BR and revegetation of BRDA.

## 2. Materials and Methods

### 2.1. BR Collection, Culture Medium and Isolation of Bacterial Strain

The aged surface BR was collected from a closed-down BRDA (in autumn 2019) of Zhengzhou, Henan province, China (113.185322° E, 34.809627° N). The collected aged BR samples were stored in plastic bags and kept at 4 °C before the isolation of the saline-alkali tolerant bacterium. The newly produced BR used for remediation was collected in the summer of 2020 and then air-dried at room temperature, ground and sieved with a 100 mesh size (150 µm).

The medium used for bacterial enrichment contained: glucose 5 g/L, yeast extract 3 g/L, KH_2_PO_4_ 0.3 g/L, MgCl_2_ 0.3 g/L and NaCl 50 g/L. The medium and NaOH solution (5 mol/L) were sterilized separately by autoclaving for 20 min at 115 °C, and then the pH of the medium was adjusted to 10.5–11.5 with the sterilized NaOH under sterile conditions. The agar plates were prepared by adding 15 g of agar to 1 L of the medium.

The enrichment procedure was similar to the description presented by Li et al. [24]. Aged BR (5 g) was added to a 250 mL Erlenmeyer flask containing 100 mL of the enrichment medium and incubated for 5 days at 30 °C on a rotary shaker (175 rpm). Next, 1 mL of the enrichment culture was serially transferred to a fresh medium three times and incubated under the same conditions. Then the enrichment was streaked onto the agar plates and incubated for two days at 30 °C. Presumptive colonies were picked on the basis of differences in colony morphology and coloration and further purified four times. Finally, seven saline-alkali-tolerant bacteria strains were successfully isolated from the BR and named ZH-1 to ZH-7. Among these, ZH-1 decreased the medium pH most significantly and was selected for further research (Appendix A).

### 2.2. Bacterial Identification and Optimization of the Medium

ZH-1 was identified using 16S rRNA gene sequencing. The universal primers (27F: 5′-AGAGTTTGATCCTGGCTCAG-3′ and 1492R: 5′-TACGGCTACCTTGTTACGAC TT-3′) were used to amplify the 16S rRNA gene. The PCR product was sequenced by the Sangon Corporation, Shanghai, China. The 16S rRNA gene sequence was compared with the known bacterial 16S rRNA gene sequences in the GenBank. The phylogenetic tree was then constructed using the neighbor-joining method with MEGA 11 software (Mega Limited, Auckland, New Zealand), and the numbers at the branch nodes are bootstrap values based on 100 re-samplings.

An orthogonal test was designed to identify the influence of glucose, yeast extract, MgSO_4_ and KH_2_PO_4_ on the change in the pH of the medium. The four factors were denoted as A, B, C and D, respectively. Three levels were selected for each factor, donated as 1, 2 and 3, respectively. The selection of these factors and levels are listed in Appendix A.

Mixed acid fermentation has been considered to be the major mechanism for the pH neutralization of BR by bacteria [10]. Therefore, the production of organic acid by ZH-1 was determined under both agitated and static conditions to explore the mechanism of pH reduction. The liquid samples were collected at 12 h intervals within 5 days for pH value and organic acid production analysis. All experiments were conducted in triplicate.

### 2.3. Experiment Design for the Remediation of BR

Two treatments were prepared based on the results of the orthogonal test, the optimal medium (without NaCl, named as CK) and optimal medium combined with ZH-1 (OD_600_ = 1.0, inoculation quantity of 5%, without NaCl, named as RX). Both were mixed with BR at a solid-to-liquid ratio of 1:2 (*w*/*v*). Then the mixture was incubated at 30 °C and weighed regularly, and the water loss due to evaporation was replenished with distilled water. The samples withdrawn at 4, 7, 11, 14, 21 and 28 days were used for pH and microbial biomass carbon (MBC) analysis. The samples collected at 14 and 28 days were used to detect the water-soluble salt content, aggregate size distribution and stability of BR. The samples collected at 28 days were used for characterization of the morphology by scanning electron microscope (SEM) and Illumina sequencing (Sangon Corporation, Shanghai, China) [25]. The total genomic DNA of the six samples collected at 28 days was extracted with Fast DNA spin kit for soil (MP Biomedicals LLC, Santa, CA, USA), and the V3 and V4 hypervariable regions of the prokaryotic 16S rRNA gene were amplified with the primers 5′-CCTACGGRRBGCASCAGKVRVGAAT-3′ and 5′-GGACTACNVGGGTWTC TAATCC-3′. All experiments were conducted in triplicate.

### 2.4. Analytical Methods

The pH of the BR was measured using a pH meter (PHSJ-3F, INESA, Shanghai, China) with a solid-to-water ratio of 1:5 (*w*/*v*). The MBC in the BR was measured according to the description given by Tao et al. [26]. The content of water-soluble salts was measured according to the standard of the ministry of agriculture of China (NY/T 1121.16-2006). The aggregate size distribution was measured by both dry and wet sieving methods, and the aggregates destruction rate (*PAD*_0.25_) was calculated according to the description given by Li et al. [27]. The content of organic acid in the solution was measured by Ultra high-performance liquid chromatography (UPLC, Waters UPLC XEVO TQD, USA) according to the description given by Ross et al. [28]. All these indicators were measured three times with a standard deviation (SD) lower than 5%.

### 2.5. Data Analysis

All data are presented as the means ± SD. The significance among different treatments were tested by ANOVA and multiple comparisons (Tukey test, *p* < 0.05) (Origin 8.5, OriginLab, Northampton, MD, USA). The correlation analyses were conducted by Origin 8.5.

## 3. Results and Discussion

### 3.1. Isolation and Identification of ZH-1

Based on the 16S rRNA gene sequence and BLAST analysis, the similarity between ZH-1 and *Bacillus* sp. KSM-K38 and *Bacillus* sp. SK2 was 99.79% and 99.77%, respectively. Figure 1 illustrates the phylogenetic relationship of ZH-1 with its close relatives. The results showed that ZH-1 is closely clustered with *Bacillus* sp. KSM-K38 with a similarity of 100%. Therefore, ZH-1 could belong to the genus of *Bacillus* sp. (GenBank accession no. MG385861). *Bacillus* sp. is widespread in different environments and has been isolated from BR as reported by Santini et al. [10]. In addition, the moderate halophilic bacteria *Nesterenkonia* sp. and *Halomonas* sp. were also isolated (Appendix A).

### 3.2. Optimization of the Medium

The results of the orthogonal experiment are presented in Appendix A. *R* represents the significance level of the factors, and *k*1 to *k*3 represents the values of levels 1 to 3. The factor with a larger *R* is more important than other factors, and the level with a larger *k* is superior to the other levels. Therefore, the order of these four influence factors on the reduction of the pH of the medium is A > D > C > B, and the optimal combination for decreasing the pH of the medium is A_3_B_2_C_1_D_1_. This result indicated that the dosage of the carbon source (glucose) is the most important factor in decreasing pH of the medium, which is consistent with the report of Santini et al. [10]. Therefore, glucose 8 g/L, yeast extract 3 g/L, MgSO_4_ 0.3 g/L and KH_2_PO_4_ 0.3 g/L was selected as the optimal medium and used for further remediation experimentation.

To explore the mechanism for pH reduction, ZH-1 was cultured under both agitated and static condition in an optimum medium (initial pH was adjusted to 11.8 with NaOH). The change of pH and the concentration of various organic acids (citric acid, oxalic acid and tartaric acid) were measured regularly. The results showed that the medium pH under both agitated and static conditions obviously decreased within 8 h of incubation, and there was no significant difference (*p* < 0.05) (Figure 2a). After that, the medium pH in the agitated condition decreased sharply to 6.14 after 48 h of incubation and then decreased gradually to a final pH of 6.01. However, the medium pH decreased gradually during the 120 h incubation under static conditions, with a final pH of 6.48. These results indicate that the culture under agitated conditions is more conducive to decreasing the medium pH, but this effect weakened with prolonged incubation time.

The production of organic acids was seriously affected by the culture condition. The concentration of citric acid under agitated and static conditions ranged from 0.35 to 0.84 mg/L and 0.39 to 1.92 mg/L, respectively, indicating that the production of citric acid by ZH-1 under the static condition is much faster than that of the agitated condition (Figure 2b). A similar result also appeared in the production of oxalic acid by ZH-1. The production of oxalic acid is fast under both the agitated and static conditions within first 12 h and then fluctuates with the concentration under the static condition slightly higher than that of the agitated condition (Figure 2c). Tartaric acid was undetectable at the initial stage, but ZH-1 began to secrete tartaric acid after 48 h (agitated culture) and 96 h (static culture) incubation (Figure 2d). At this time, the medium pH was 6.14 and 7.42, respectively, indicating that ZH-1 secretes tartaric acid only under acidic or neutral conditions. Correlation analysis indicated that the medium pH was significantly negatively correlated with the secretion of total organic acid content (Appendix A). These results further verified that acid fermentation was the major mechanism for decreasing the pH of the medium, which is consistent with previous reports [20,29].

### 3.3. Neutralization of BR

The changes in the pH value of BR are shown in Figure 3. In the CK treatment, the pH value of BR decreased from 11.6 to 10.8 within 4 days and then further decreased with incubation time, reaching the lowest pH value of 10.5. In the RX treatment, the pH value of BR decreased from 11.6 to 9.7 within 4 days and then decreased gradually to 8.8 after 28 days of incubation. Due to the low pH of the yeast extract, the pH of the optimal medium is about 5.5 in the present study. Therefore, the pH of the BR in both the CK and RX treatments were decreased obviously after mixed with the optimal medium (0 to 4 days), but the RX treatment decreased more obviously, which indicates that ZH-1 promoted the decrease in the pH of the BR.

High pH presents a great challenge for the revegetation of BRDA, and various amendments have been used to neutralize the alkalinity of BR in previous studies, such as gypsum [30], carbon dioxide [31], acidic fly ash [18], seawater [32], organic carbon [33] and livestock manures [34]. In addition, the application of organic amendments can enhance the number and metabolic activity of indigenous microorganisms [33,35] and decrease the pH of BR. However, most previous studies emphasized the importance of amendments, and the roles of microorganisms in reducing the pH of BR was ignored to a certain extent. With the rapid development of biotechnology, the critical role of microorganisms in decreasing the pH of BR has been recognized [20,29]. Some exploratory studies on reducing the pH of BR by microorganisms have been conducted in recent years, including bacterial, fungal and microbial-rich amendments [33,35]. However, insufficient attention has been paid to the influence of microorganisms on other properties of BR (except for pH).

### 3.4. Change in Water-Soluble Salt Content

After 14 and 28 days of incubation, the water-soluble salt content increased from 7.84 to 8.32 and 8.56 g/kg in the CK treatment and from 7.84 to 10.25 and 13.18 g/kg in the RX treatment (Figure 4a). The application of the medium increased the salt solubility slightly, but the change between 14 and 28 days was not significant. ZH-1 inoculation increased the salt solubility by 23.20% and 53.97% compared with CK after 14 and 28 days of incubation, respectively. In addition, the change of water-soluble Na^+^ was similar to that of the water-soluble salt (Figure 4b). Water-soluble Na^+^ increased slightly with the incubation time in the CK treatment. However, its content increased more than 54.59% and 78.75% after 14 and 28 days of incubation, respectively, in the RX treatment. These results indicate that ZH-1 can enhance the release of salt in BR.

High salinity is another challenge for the revegetation of BRDA, which can inhibit seed germination, water and nutrient elements absorption, and photosynthesis of plants [36,37]. Various methods have been used to reduce the salt content in BR, including water leaching, roasting [38], acid transformation [15] and chloride salt neutralization [39]. However, these technologies are expensive or have potential secondary pollution as compared with bioremediation. ZH-1 application not only decreased the pH of BR but also significantly enhanced the release of salt and Na^+^ (the most representative salt ion) from the BR. Furthermore, ZH-1 was isolated from the BR, and its potential ecological risk is controllable. These results indicate that ZH-1 may have great application potential in the revegetation of BRDA.

### 3.5. Change in Aggregate Distribution

Aggregate is a basic physical property of soil, which can reflect the water retention, permeability, and erodibility of soil and soil analogs [34]. The aggregate size distribution of BR in the CK and RX treatments are presented in Figure 5. For the dry sieving method, the aggregate size increased obviously in both treatments. The mechanical-stable aggregates (D_0.25_) increased from 69.11% to 80.23% (14 d) and 87.87% (28 d) in CK, and to 93.23% (14 d) and 98.55% (28 d) in RX. A similar result appeared for the wet sieving method, the water-stable aggregates (W0.25) increased from 3.89% to 9.38% (14 d) and 14.27% (28 d) in CK, and to 24.74% (14 d) and 18.82% (28 d) in RX.

The addition of the medium (CK) can stimulate the growth of indigenous microorganisms, and enhance the aggregate formation of BR by extracellular polymeric substance and fungal hyphae [40]. At the same time, the application of ZH-1 (RX) further promoted the formation of aggregates compared with CK, indicating that the existence of the saline-alkali tolerant bacteria is conducive to the formation of aggregates. SEM analysis showed that the BR in CK treatment contained many small flake-structured, irregular shapes and unsmooth edge particles (Figure 6a,b). In RX treatment, the flake structure became larger, regular, and compact with a flat edge, and the size of the aggregates obviously increased (Figure 6c,d). These results further verified that ZH-1 improved the physical properties of BR.

### 3.6. Change in Aggregate Stability

The stability of the aggregates were assessed by the percentage of aggregate destruction (*PAD*_0.25_), which was calculated using the following equation:(1)$$PAD_{0.25}=\frac{D_{0.25}-W_{0.25}}{D_{0.25}}\times 100\%$$
where *D*_0.25_ and *W*_0.25_ are the percentage of mechanical-stable aggregates and water-stable aggregates, respectively.

The influence of ZH-1 on aggregate stability is presented in Appendix A. The stability increased significantly (*p* < 0.05) with the addition of the medium, as the *PAD*_0.25_ decreased from 94.37% to 88.31% and 83.76% at 14 and 28 days, respectively. Previous reports indicated that organic carbon content was positively correlated with the stability of aggregates and the addition of organic carbon can improve the stability of both macro- and micro-aggregates, which are consistent with the result of the present study [41,42]. The *PAD*_0.25_ further decreased with the inoculation of ZH-1, which decreased to 73.46% and 77.90% at 14 and 28 days, respectively. Although *PAD*_0.25_ slightly increased at 28 days, but the difference was not significant. These results indicated that ZH-1 inoculation made the aggregates more stable and less likely to be destructed.

Various organic amendments have been used to improve the aggregate stability of BR, such as vermicompost [43], humic acid, humic acid-acrylamide polymer, straw [44], and spent mushroom compost [45]. However, few studies regard the influence of microorganisms on the formation and stability of BR aggregate have been reported previously. In addition, the formation of aggregates and improvement of the physical properties of BR is conducive to plant colonization and growth [46,47]. These results further verified the application potential of ZH-1 in BRDA revegetation.

### 3.7. Change in MBC and Bacterial Community of BR

The total content of microbes in the original BR is low (425 mg/kg) due to its extreme environment (Figure 7). Some indigenous microorganisms can survive and recovery in the BR after the addition of the medium, but the growth rate is slow. The MBC in CK increased slowly and finally reached 978 mg/kg after 28 days of incubation. Whereas the MBC in RX increased to 2794 mg/kg after 28 days of incubation, which is nearly three times higher than that of the CK. This result indicates that ZH-1 can adapt to the BR environment and grow rapidly. Correlation analysis also showed that the pH value of BR is significantly negatively correlated with the MBC in the RX treatment (*p* < 0.01, Appendix A), whereas the correlation is not significant in that of the CK treatment (Appendix A). Therefore, we concluded that the decrease of the pH value in BR was mainly driven by ZH-1. These results are partially consistent with the report of Santini et al. [33], who reported that just the application of organic carbon can significantly reduce the pH of BR. 

With the consumption of the added organic carbon, the metabolic activities of the microorganisms in BR decreased gradually, and some of them may even enter a dormant state. At this time, their influence on the properties of BR is weakened. However, when new carbon sources (plant litter, straw, animal manure, leaves, etc.) entered into the BR, the microorganisms will recover, which can further improve the properties of the BR. Therefore, the enhancement of MBC can realize the long-term improvement of BR, which can further improve the colonization and growth of plant. 

The bacterial diversity and community compositions in neutralized BR at class level are shown in Figure 8. Due to the fact that total DNA cannot be extracted from the original BR, the bacterial diversity and community composition cannot be shown in the present study. Bacterial community analysis showed that approximately thirteen known classes were identified from the neutralized BR in both CK and RX treatment, and more than 98% of these classes were affiliated to Clostridia, Bacilli, Gammaproteobacteria, Betaproteobacteria and Alphaproteobacteria (Figure 8). Bacilli, Gammaproteobacteria and Alphaproteobacteria have been identified from natural weathered BR by Wu et al. [48], but the classes of Clostridia and Betaproteobacteria have not been identified from BR in previous reports. Krishna et al. reported that Betaproteobacteria, Gammaproteobacteria and Bacteroidetes are the major bacteria in BR [49], which is partially consistent with our present study.

Although the dominated bacterial classes in these two treatments were similar, the inoculation of ZH-1 still affected the bacterial community in BR. The relative abundance of Bacilli (ZH-1 belonged to the class of Bacilli) in RX (42.64%) was much higher than that of CK (28.51%), the relative abundance of Gammaproteobacteria in RX (1.61%) was much lower than that of CK (26.04%). In addition, the alpha diversity analysis revealed that the addition of ZH-1 increased the richness and evenness of bacterial communities (Ace, Chao1, Shannon and Simpson) after 28 days incubation (Table 1). The Ace, Chao1, Shannon and Simpson indexes increased from 67.20, 64.67, 3.77 and 0.88 in the CK treatment to 88.81, 88.00, 4.14 and 0.91 in the RX treatment, respectively. These results indicate that ZH-1 not only increased the total microorganism content, but also enhanced the diversity of microorganisms in the neutralized BR. Previous reports indicated that biodiversity is very important in maintaining the ecological process [50,51,52] indicating that ZH-1 can play an important role in the revegetation of BRDA.

## 4. Conclusions

The saline-alkali tolerant bacteria ZH-1 (*Bacillus* sp.) was successfully isolated from BRDA and identified. ZH-1 could secrete various organic acids under optimum medium conditions and decrease the pH of the medium. ZH-1 could also neutralize the alkaline and enhance the salt release of BR. In addition, the aggregate size distribution and stability of BR was significantly improved by ZH-1, and the MBC content and bacterial diversity in BR were also significantly increased. The results of present study suggest that ZH-1 could be a potential candidate for improvement of BR and may be beneficial for ecological reconstruction in BRDA.

## Figures and Tables

**Figure 1 ijerph-19-11590-f001:**
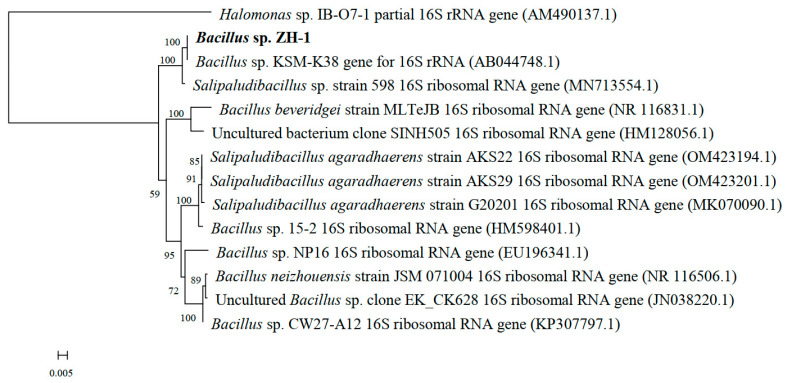
16S rRNA gene sequence-generated phylogenetic tree showing the relationships of *Bacillus* sp. ZH-1 and its closest relatives.

**Figure 2 ijerph-19-11590-f002:**
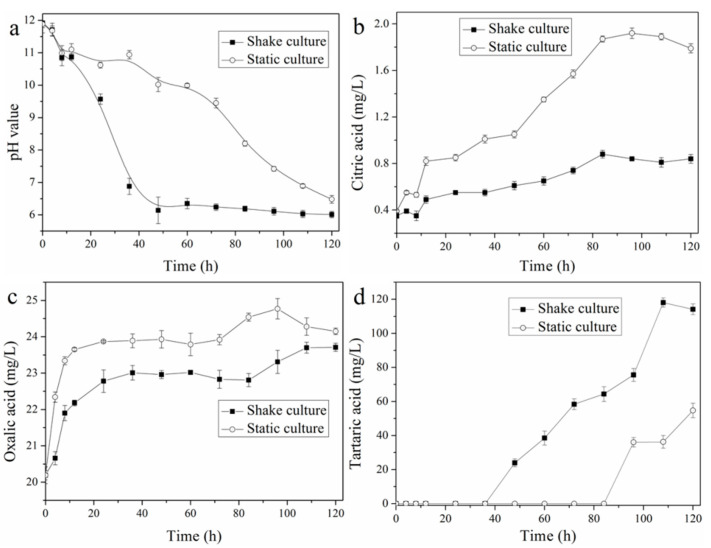
Change in pH value and production of organic acid by ZH-1 in optimal medium under both agitated and static conditions. (**a**) Change in pH; (**b**) production of citric acid; (**c**) production of oxalic acid; (**d**) production of tartaric acid.

**Figure 3 ijerph-19-11590-f003:**
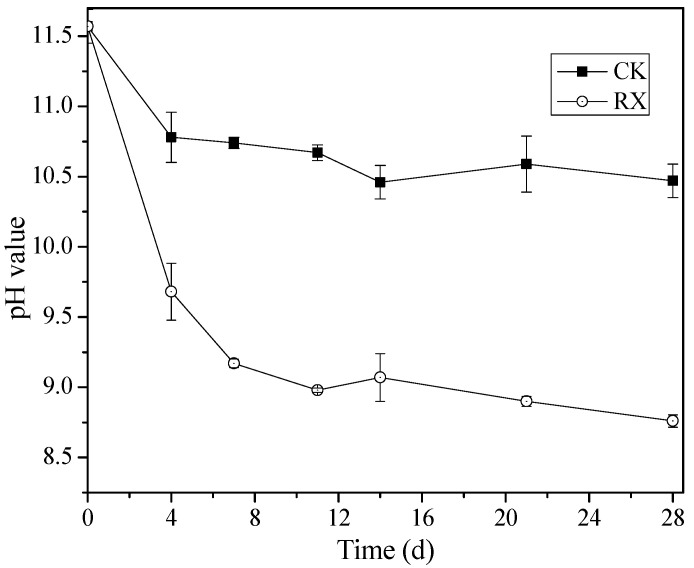
Change in pH value of BR with (RX) and without (CK) the application of ZH-1. These results are presented as mean ± standard deviation (*n* = 3).

**Figure 4 ijerph-19-11590-f004:**
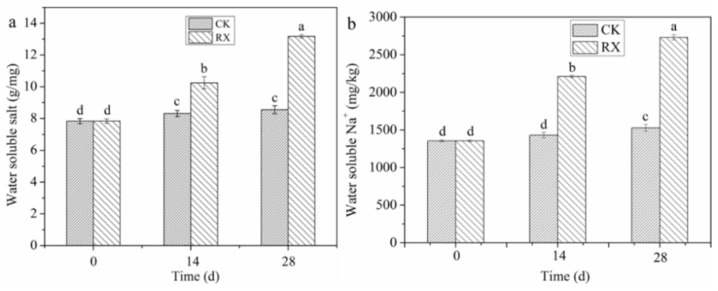
Change in water-soluble salt and water-soluble Na^+^ content after the application of ZH-1. (**a**) Water soluble salt; (**b**) water soluble Na^+^. The different lowercase letters above the column indicate the water-soluble salt and Na^+^ with significant differences (ANOVA, Tukey test, *p* < 0.05).

**Figure 5 ijerph-19-11590-f005:**
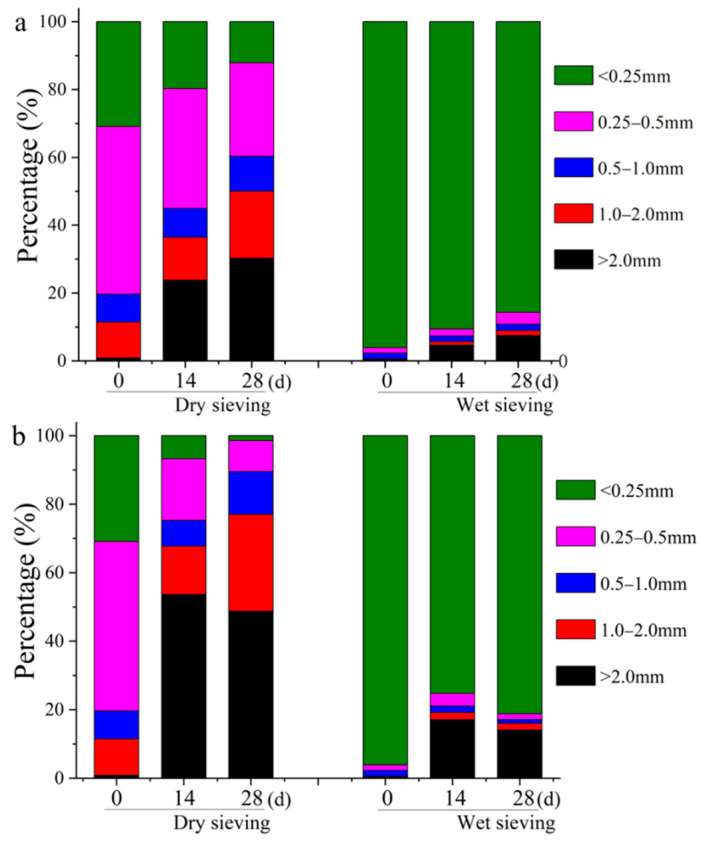
Aggregate size distribution of BR in CK (**a**) without the application of ZH-1; and RX (**b**) application of ZH-1 treatment (*n* = 3).

**Figure 6 ijerph-19-11590-f006:**
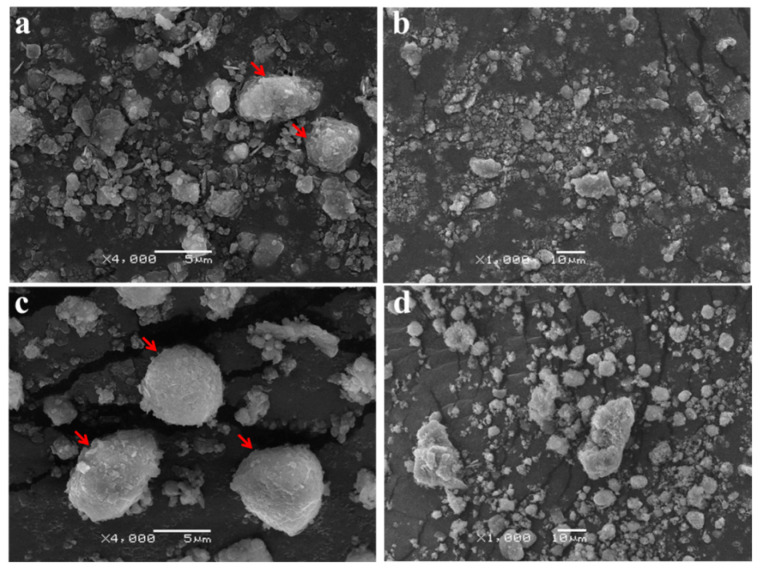
SEM image of BR in CK (**a**,**b**), without the application of ZH-1; and RX (**c**,**d**), with the application of ZH-1.

**Figure 7 ijerph-19-11590-f007:**
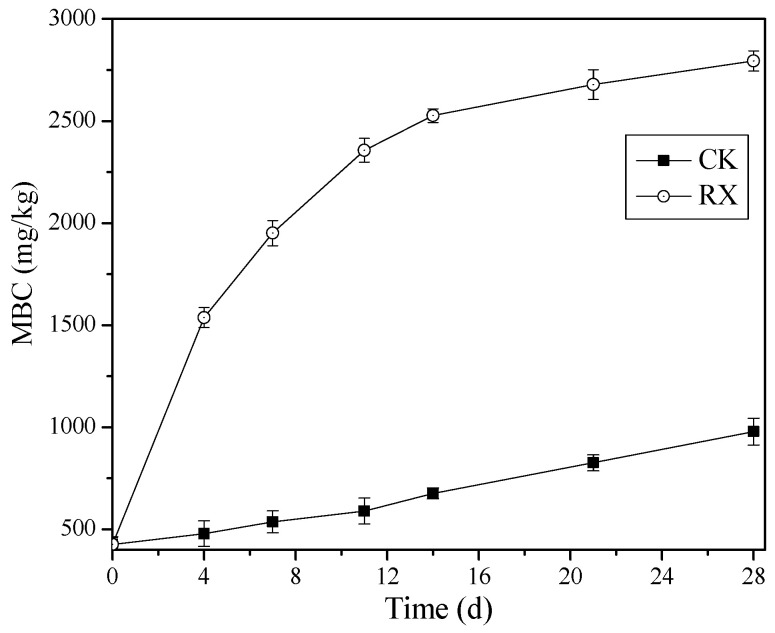
Change of MBC in BR with (RX) and without (CK) the application of ZH-1. These results are presented as mean ± standard deviation (*n* = 3).

**Figure 8 ijerph-19-11590-f008:**
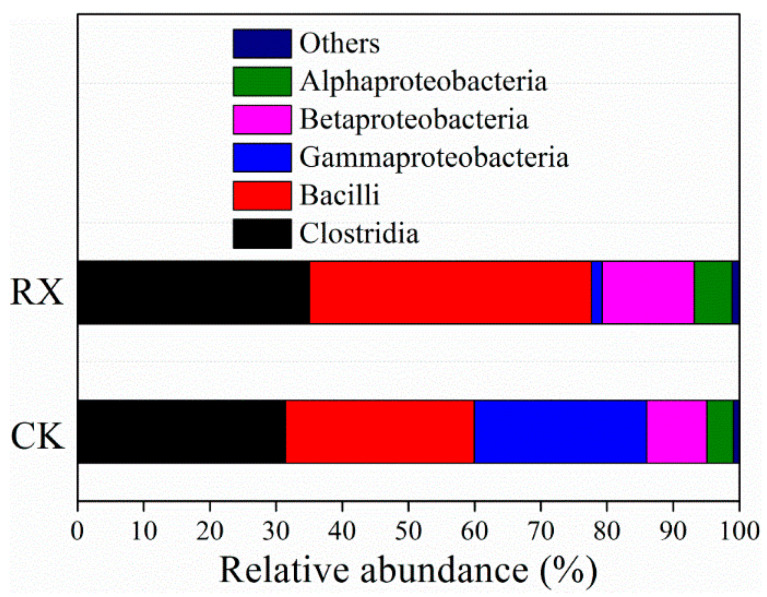
Bacterial community in neutralized BR by ZH-1 at class level.

**Table 1 ijerph-19-11590-t001:** Summary of Illumina Miseq high-throughput sequencing of 16S rRNA gene.

Sample	Ace	Chao1	Shannon	Simpson	Coverage
CK	67.20	64.67	3.77	0.88	1
RX	88.81	88.00	4.14	0.91	1

## Data Availability

Not applicable.

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
