# Peer review of "Neutralization and Improvement of Bauxite Residue by Saline-Alkali Tolerant Bacteria"

_ijerph, 2022, doi:10.3390/ijerph191811590_

Round 1
Reviewer 1 Report
Comment on
Neutralization and Improvement of Bauxite Residue by Newly Isolated Saline-Alkali Tolerant Bacteria and Implications for Revegetation
Why bauxite is focused in this study? Is there any toxicological concern to the environmental and what is the human health risks to bauxide? Please justify.
When the samples were collected?
Please include QA and QC in the analytical methods.
Please include why and how did you select the bacteria strains? Any criteria followed?
What is the natural pH level in the bauxite residue disposal areas (of your study area)? What is the degree of effectiveness of this method if your laboratory results ar applied in the field? Do you think any field trials need to be conducted and why. Please explain and discuss.
Please check English again.
Reviewer 2 Report
The work presented by Lv and collaborators is entitled Neutralization and Improvement of Bauxite Residue by Newly Isolated Saline-Alkali Tolerant Bacteria and Implications for Revegetation, reports the discovery of a new strain identified as Bacillus sp and that is capable of neutralizing bauxite residues, changing the size of the aggregate distribution and its mechanical stability and the stability of aggregates in water. The impact of the application of this bacterium on sites contaminated with bauxite could favor their revegetation.
Major Comments
1) The first thing that stands out is that the authors refer to a new strain when it has 100% genomic identity of the 16S rRNA with the alkaliphilic Bacillus isolate KSM-K38 strain. This could indicate that it could be the same strain and it cannot necessarily be said that it is a new one.
Regarding this point, the methodology used to make the mini phylogenetic tree is not described: what primers were used for 16S rRNA gene amplification, what algorithm was used (for example, Maximun Likelihood, Neighbor-Joining), nor does it indicate the number of bootstraps, or the software used for these purposes?
2) The second thing that stands out is that the title mentions revegetation as the final objective, which apart from the title is mentioned 9 times in 7 pages. However, this manuscript does not prove that the neutralization by the bacterial isolate has any kind of implication on revegetation, since there are no experiments, whose results demonstrate this. Indeed, ¨revegetation¨ it is not present in conclusions. I strongly suggest changing the title to an adapted version of the purpose of this manuscript.
3) Regarding the results obtained from the relative abundance obtained from the microbial communities (Fig 8), it is not indicated in materials and methods how the DNA was extracted, which variable region was amplified, how many samples were run, etc.
4) Nowhere is it explained why it was decided to determine two intermediate acids of the tricarboxylic acid cycle (citric and oxalic acid) and one peripheral acid (tartaric acid).
5) The initial culture medium contains Glucose 5 g/l and NaCl 50 g/l. The optimized medium contains Glucose 8 g/l and it is not indicated if the NaCl concentration was modified. Indicate what was the criterion for raising the glucose concentration from 5 to 8 g/l and not to another value and if the NaCl concentration was maintained.
6) Add a field in Fig. 6 at a lower zoom to see a more complete field.
Minor Comment
Line 25 Bacillus in italic
Reviewer 3 Report
Dear authors,
I carefully checked the MS called « Neutralization and improvement of bauxite residue by newly isolated satline-alkali bacteria... » by Lv et al, which focused on the biological abilities to one Bacillus species to modify the pH and improve the other properties of bauxite residues in two media in a perspective of bioremediation of polluted environements. The paper is well writing, with a very concise and comprehensive introduction followed by some precise and informative results & discussion section as well. Despite this, some improvements can be made in each title of the figure and in the figures as well, (sometimes incomplete, or not easily visible) for a better understanding and for increasing the clarity of the whole paper.
Fig1 : The tree only based on NJ method is insufficient, while the actual consensus for the phylogenetic analyses combined at least 2 methods, ie NJ and ML or Bayesian construction. Besides, it is surprising to have only few sequences including Bacillus genus to compare with your strain. The phylogenetic part must be completed and more justified. How many boostraps here ? Why not including the % values instead? Consequently, the additional data must be included in the Mat & Meth section.
The title : ...and its close relatives... » or closest relatives ?
Fig 2 : the symbols are too close to really see who is who. Try an « empty » vs a « full » symbol maybe, or a different color… These graphs show the modifications when using the bacteria, right ? It is not mentioned in the title (the same for other figures as well). A control is expected, without the bacterial addition, to really show that any modification of pH and other properties occured naturally with time incubation, isn’t it ?
Fig3 : the title is not so informative nor precise. Please add some details, i e with or without bacteria ? What is the n number ? Replicates ? standard deviation ? Remind the abbreviation used (what’s CK and RX meaning…)
nonsignificant= better « no significant » 2 words
Fig 4 : Title : the letter in each graph is missing. They are supposed to validate significant vs no significant differences right ? This information must be included in the title as well (with the statistical test used here). In the text the authors detected no significant (lines 219-224), while some differences seem to be occurred at different times...What statistical test is used here ?
Fig 5 : title must be more detailed as well (see above).How many replicates here in dry sieving vs wet sieving ?
Same remarks for the title in Fig 6. The scale bars meaning ? Where are the bacteria on or around the aggregates ? Can we see them or it is not possible in a & b ? Is it a SEM micrograph ? All these informations must be noticied to this figure.
In the formulation PAD= what is the « P » meaning ?
Fig 7 : same remarks as above. Change the symbols (too close and not clearly different!) and complete the title as well. ..Remind the abb « MBC » also…
- The last part of the results/discussion section below the fig 6 could be presented by a specific subtitle here.
- To highlight the results obtained in the MS, maybe it could be appropriate to include a table with all significant vs no significant correlations between all the treatments and physical properties of media improved by the addition of bacterial strain ? Just to better focus and synthesise the modifications occuring and the bioremediation potential of bacteria strain for BR medium .
- About the form and typological errors :
At last, the authors must re-read their MS because a lot of typological and formal errors exist throughout the paper. For example microorganisms in general is plural with an « s » and « have » and not has… »inoculum »(line 62-63) inappropriate when referring to several... some words are missing (line 260 : « which calculated using.. » « was « is missing for was calculated...and so on… the text must be re-read with attention.
Round 2
Reviewer 2 Report
Authors have made significant changes to the text and have added the requested information. Therefore, this manuscript is suitable for publication.